# Quasi-seamless stitching for large-area micropatterned surfaces enabled by Fourier spectral analysis of moiré patterns

Woo Young Kim[1], Bo Wook Seo[1], Sang Hoon Lee[1], Tae Gyung Lee[1], Sin Kwon[2], Won Seok Chang[3,4], Sang-Hoon Nam[5], Nicholas X. Fang [5,6], Seok Kim [1,7] ✉ & Young Tae Cho [1,7] ✉

The main challenge in preparing a flexible mold stamp using roll-to-roll nanoimprint lithography is to simultaneously increase the imprintable area with a minimized perceptible seam. However, the current methods for stitching multiple small molds to fabricate large-area molds and functional surfaces typically rely on the alignment mark, which inevitably produces a clear alignment mark and stitched seam. In this study, we propose a markless alignment by the pattern itself method inspired by moiré technique, which uses the Fourier spectral analysis of moiré patterns formed by superposed identical patterns for alignment. This method is capable of fabricating scalable functional surfaces and imprint molds with quasi-seamless and alignment mark-free patterning. By harnessing the rotational invariance property in the Fourier transform, our approach is confirmed to be a simple and efficient method for extracting the rotational and translational offsets in overlapped periodic or nonperiodic patterns with a minimized stitched region, thereby allowing for the large-area and quasi-seamless fabrication of imprinting molds and functional surfaces, such as liquid-repellent film and micro-optical sheets, that surpass the conventional alignment and stitching limits and potentially expand their application in producing large-area metasurfaces.

The increasing demand for nanopatterned functional surfaces in numerous emerging applications, such as optical[1,2], biological[3-7], energy harvesting[8], lithographic systems[9], and environmental devices[10], has stimulated thelarge-area micro-and need for scalable manufacturing technologies[2,11-13]. Functional surfaces are commonly composed of a two-dimensional (2D) or quasi-three-dimensional (3D) array of micro- and nano-scale structures with periodic or aperiodic arrangements[11,13,14]. As these arrayed structures must have precise dimensions and arrangements to realize high performance according to specific design intents in the required surface area[2], an elaborate fabrication technique is essential. Accordingly, roll-to-roll[15-17] or roll-to-plate[18-20] nanoimprint lithography (R2R-NIL or R2P-NIL, respectively), a mechanical patterning method, is an irreplaceable manufacturing tool because it is capable of the continuous and cost-effective production

[1]Department of Smart Manufacturing Engineering, Changwon National University, Changwon, South Korea. [2]Department of Flexible & Printed Electronics, Korea Institute of Machinery and Materials, Daejeon, South Korea. [3]Department of Nano Manufacturing Technology, Korea Institute of Machinery and Materials, Daejeon, South Korea. [4]Department of Nanomechatronics, University of Science and Technology, Daejeon, South Korea. [5]Department of Mechanical Engineering, Massachusetts Institute of Technology, Cambridge, MA, USA. [6]Department of Mechanical Engineering, The University of Hong Kong, Hong Kong, Hong Kong, Special Administrative Region of China. [7]Department of Mechanical Engineering, Changwon National University, Changwon, South Korea. ✉e-mail: kimseok@changwon.ac.kr; ytcho@changwon.ac.kr

of multiscale structures over square-meter-large and flexible substrates[9,11,12,21–24]. However, the challenges of preparing large-area imprint molds significantly limit the critical advantage of roll-based NIL for the large-area production of functional surfaces[23–26]. Because the original master molds are typically fabricated using e-beam lithography, photolithography, or interference lithography, the small mold should be repeatedly stitched to create a flexible large replicate mold to sufficiently cover the circumference of the roll, resulting in unavoidable seam regions. Accordingly, extensive efforts have been made to fabricate scalable and flexible imprint molds[23–26]. Owing to these difficulties in fabricating multiscale patterns directly over a large area, a step-and-repeat process using mechanical alignment system[26,27] and shadow mask has been developed to fabricate large-area master molds[9,24,26,28]. Although multiple-stitched flexible molds have been widely used in roll-based NIL, there is a critical limit to satisfying the requirements of practical applications because of the clear formation of alignment marks and stitching seams, which are easily detectable by the naked eye[23,29]. As alternatives, novel mechanical approaches, such as dynamic nanoinscribing[25], nanochannel-guided lithography[30], and vibrational indentation-driven patterning[31], have been proposed to achieve scalable seamless and alignment mark-free patterning. Inspired by the traditional forging and indentation processes, these patterning methods mechanically deform substrate materials and carve periodic patterns on the substrates through the continuous sliding motion or vibration of hard molds. Although these methods enable the fabrication of continuous and seamless large-area patterns, they are still limited to producing multiscale complex patterns except for monotonous shapes, such as gratings, owing to their simple mechanical mechanism. An overlapped tiling method that does not require the use of alignment marks, such as visually tolerable tiling or transfer tiling[23,32], has recently been developed to manufacture large-area and quasi-seamless molds using overlapping small stamps. In these tiling methods, the small master pattern is slightly overlapped by harnessing the dewetting property of the photocurable resin, thereby demonstrating visually tolerable seam regions. Because it is unrealistic to achieve perfect alignment at the nanoscale, considering cost-effectiveness and commercialized equipment, they can be one promising strategy to achieve practical large-area imprint molds. Accordingly, a systematic and precise arrangement method for arrayed patterns is required to maximize the effectiveness of consecutive overlapping or stitching of small stamps for preparing quasi-seamless large-area imprint molds. In this study, a novel stitching method based on mark-less alignment by the pattern itself (MAPI) is proposed for large-area micropatterned surfaces using the Fourier spectral analysis of two overlapped identical patterns during the imprint stitching process. The central idea is that the spectral analysis of the moiré pattern of entangled identical patterns can replace the functionality of a typical alignment mark. The moiré patterns produced by two periodic or aperiodic structures placed in contact or projected onto another have been used for the analysis of translation and angular displacement[13,33–37]. By analyzing the moiré patterns of the superposed region between the imprinted patterns and transparent mold, the overlapping regions were extracted to compensate for rotational and translational misalignment. Because real-space patterns can be described as $k$-space patterns through the Fourier transform (FT), the angular misalignment can be extracted from the $k$-space images of the measured real-space images by harnessing the rotational invariance property in FT[38–40]. Subsequently, the translational misalignment can be measured using vision-based shifted pixel analyses. Based on the extracted rotational and translation errors, the alignment of the imprint mold was adjusted using a precise motorized stage, and then it was stitched to fabricate large-area micropatterned molds using a step-and-repeat process. In addition, rectangular tiling using the shadow mask method supports seamless stitching in a plane. To demonstrate the effectiveness of the proposed stitching method for alignment mark-free patterning, large-area quasi-seamless functional surfaces, including a hydrophobic film and microlens array sheet, were fabricated.

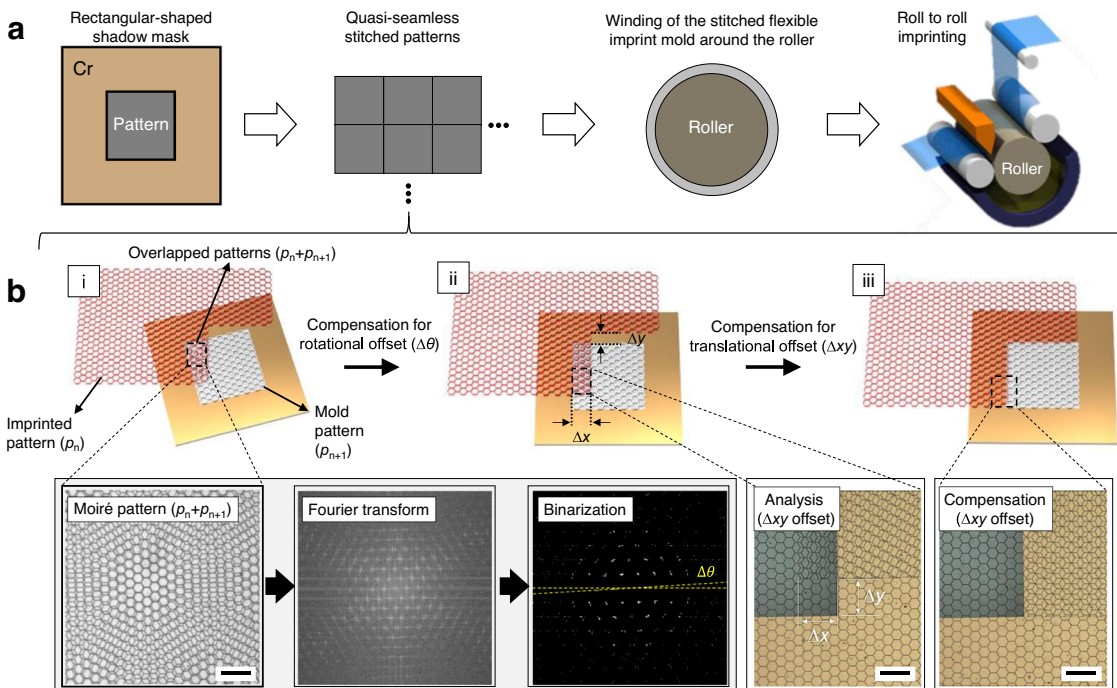

**Fig. 1 | Overview of the MAPI method. a** Overall schematic of the R2R NIL process using a quasi-seamless flexible mold. **b** Schematic of the MAPI method in the overlapped area between the imprinted pattern ($p_n$) and imprint mold pattern ($p_{n+1}$). (i) Inevitable rotational and translation offsets occurs in the overlapped area due to an initial random alignment state; (ii) remaining translational offset after rotational alignment based on the moiré pattern analysis; and (iii) the quasi-seamless tiled patterns after compensating for the translational offset. Scale bar: 200 μm.

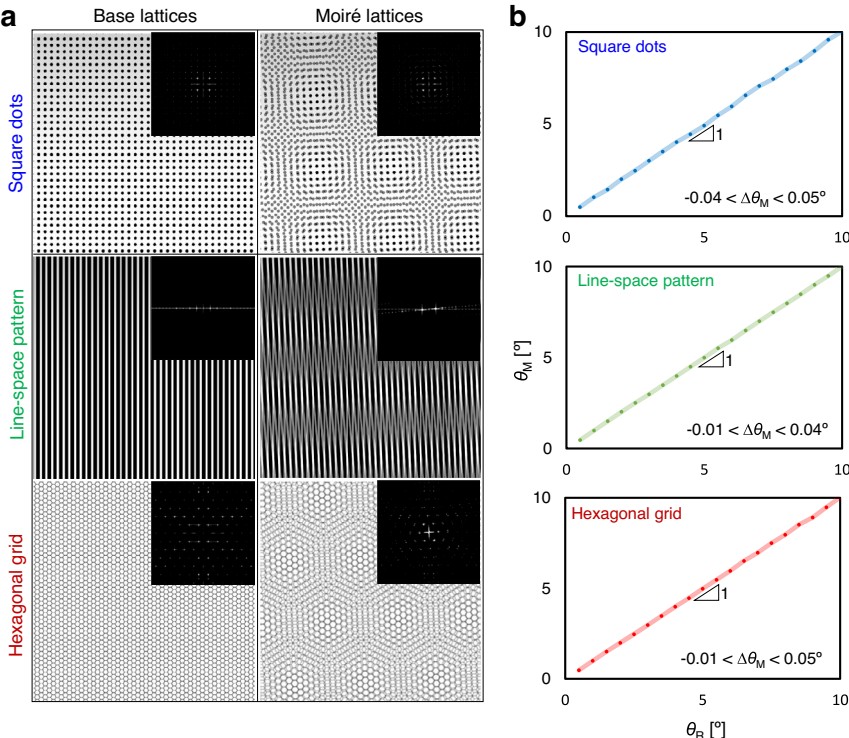

**Fig. 2 | Simulated results for the extraction of the rotational offset using a Fourier spectral analysis of the moiré lattices formed by two identical overlapped patterns. a** Simulated images of base and moiré lattices (insets show the corresponding Fourier-transformed images of the overlapped patterns ($p_n + p_{n+1}$) for the periodic structures). **b** The measured rotation angle ($\theta_M$) as a function of the input rotation angle ($\theta_R$).

## Results and discussion

### Description of the MAPI method using Fourier spectral analysis of moiré patterns

Figure 1a schematically illustrates the R2R-NIL process for the large-scale fabrication of functional films through a quasi-seamless flexible imprint mold tiled using the proposed MAPI process. Figure 1b illustrates the step-and-repeat stitching method based on the MAPI of small-area master pattern stamps to produce large-area imprint molds. In this tiling process, a small master stamp containing the desired multiscale pattern is placed on top of the substrate, and the photocurable resin is dispensed. After imprinting the $n$th pattern ($p_n$), the subsequent $n + 1$th pattern ($p_{n+1}$) is slightly overlapped and tiled by the stitching process, repeating itself to fabricate large-area imprint molds. The overlapped area defined as $p_n + p_{n+1}$ has a clear seam owing to the relative rotational and translational misalignment between two identical periodic patterns, thereby resulting in moiré patterns with high rotational symmetry as shown in Fig. 1b (i). Because of the ultraviolet (UV) exposure-based NIL, the moiré structures formed by the overlapped patterns in the transparent samples can be directly measured. Figure 1b shows that these real-space moiré structures can be converted to the reciprocal lattice ($k$-space) by FT and be presented as the twist and shift of the lattice in the $k$-space. The relative in-plane rotation angle ($\theta$) between the top layer (imprinted pattern, red) and bottom layer (imprint mold, black) can be extracted from the $k$-space image by harnessing the rotational invariant property of FT. The precise mechanical rotation stage first controls the top layer to compensate for the misaligned angular offset, leaving behind the translation misalignment (Fig. 1b (ii)). The spatial offset for translational misalignment can then be extracted by analyzing the shifted pixels between two identical periodic patterns and compensated for by controlling the motorized translational stage, leading to quasi-seamless tiling (Fig. 1b (iii)). Considering the irregular gap between the already imprinted substrates and the mold to be subsequently

imprinted, moiré pattern analysis is suited to the alignment for the fabrication of large-area imprint molds with a step-and-repeat stitching process because the moiré patterns are independent of the gap between the two periodic patterns[41]. According to the rotation theorem of the FT operator, any rotation of the original images in real-space corresponds to the rotated reciprocal images in $k$-space, which is called rotational invariance. For example, the FT image of the hexagonal patterns ($p_n$ and $p_{n+1}$) appears as a shape rotated in the direction in which the real pattern is rotated as shown in Fig. 1b. Accordingly, the FT image of the superposed patterns ($p_n + p_{n+1}$) results from the linear combination of each basis vector in the $k$-space with the same reciprocal lattice constant but with a rotation offset ($\Delta\theta$) between them.

### Demonstration of the MAPI process

To verify the rotational misalignment measurement using Fourier spectral analysis, moiré structures were simulated with varying angular offsets between two identical overlapping lattices composed of square dots, line-space patterns, and hexagonal grid patterns (Fig. 2a). 2D modeling images (resolution: 600 × 600 pixels) of periodic and aperiodic patterns were created using AutoCAD, which were overlapped by varying the rotation angle from 0 to 10° (0.5° intervals) to simulate real-world conditions in actual applications, thus yielding the moiré patterns. The modeling images of moiré patterns were first converted to the corresponding reciprocal images using FT, and then the specific nodes (bright spots) in the Fourier-transformed images were analyzed, which were determined by their periodicity and angular offsets. The FT images were binarized to extract these specific node points more clearly (insets in Fig. 2a). Three reference and three rotated diffraction spots in the binarized images, which were located at the farthest points from the center point, were extracted and the angular rotation was calculated accordingly (please see the details in Supplementary Fig. 1). It was noted in these simulation results (Fig. 2b) that the measured

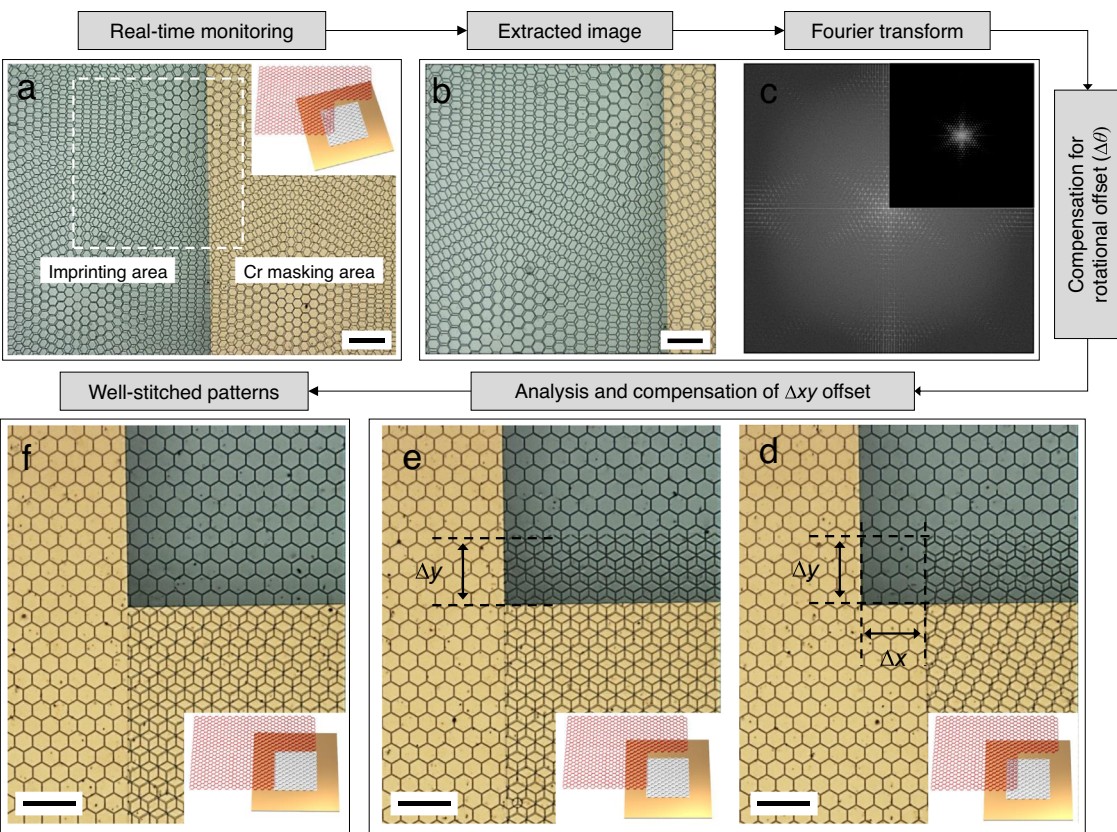

**Fig. 3 | Sequential explanation of the proposed MAPI process. a** Real-time monitoring of the moiré patterns in the superposed area. **b** Extracted image for FT. **c** Analysis and compensation for the rotational offset ($\Delta\theta$). **d,e** Analysis and compensation for the translational offset ($\Delta xy$). **f** Quasi-seamless micropatterned surfaces after completion of the MAPI process. Scale bar: 200 μm.

rotation angle ($\theta_M$) exhibited a linear relationship with the input rotation angle ($\theta_R$) with a constant of 1 and a reasonable deviation below 0.05°, originating from the simulated image resolution. This strongly supports our hypothesis that the moiré pattern analysis using FT can accurately estimate the rotational offset of overlapped patterns. It should be noted that the mechanism was also applicable to more comprehensive patterns, such as aperiodic patterns of Penrose tiling and classical patterns for metasurfaces (Supplementary Fig. 2), and potentially expand their capability in producing broad functional surfaces.

A custom-designed printing system was built as a proof-of-concept (Supplementary Fig. 3), and step-and-repeat stitching was performed based on the proposed MAPI process (Fig. 3). It should be noted that chrome-coated glass with a rectangular-shaped transparent pattern (Supplementary Fig. 4a) was used as the shadow mask to provide seamless stitching in the translational stitching process[25]. This transparent and rectangular shadow mask mold helps to monitor and control the MAPI process in the uncured photopolymer underneath. A photocurable polyurethane acrylate-based resin was first dropped next to the already imprinted patterns ($p_n$) and then was gently pressed with a shadow-masking glass stamp containing an imprinting structure of hexagonal interconnected microcavities. In this study, an interconnected microcavity pattern with a periodic hexagonal arrangement was used as a model structure to demonstrate the MAPI process based on moiré pattern analyses. The shadow mask was first positioned and aligned using the MAPI process; then, the next pattern ($p_{n+1}$) was imprinted using UV exposure, followed by demolding of the shadow mask (Fig. 3a). The optical image of the overlapped patterns was then captured and the corresponding FT image was analyzed (Fig. 3b, c), resulting in the estimated angular offset ($\Delta\theta$). The top imprinted pattern was rotated by as much as the estimated offset while

fixing the bottom pattern. After aligning the rotational offset, the translation offset was compensated for using a vision-based shifted pixel analysis, as shown in Fig. 3d–f (please see the details in Supplementary Fig. 5). The inevitable large offset (scale of hundreds of micrometers) originating from the initial random orientation state was easily analyzed using the measured optical images and subsequently compensated for using the precise motorized positioning system. Accordingly, the two identical patterns were experimentally confirmed to be rotationally and translationally well-aligned, as shown in Fig. 3f.

**Evaluation of the MAPI-based stitching process**

To quantitatively study the performance of this MAPI-based stitching process, the angular ($\Delta\theta$), translational ($\Delta seam$ and $\Delta edge$), and height ($\Delta h$) offsets of the stitched areas were analyzed and compared with those using the manual stitching process (Fig. 4 and Supplementary Fig. 6). Optical and 3D profile images of the microstructured surfaces were obtained using a laser scanning confocal microscope (Keyence, VK-X1000) to analyze these offsets. Two types of representative microstructured surfaces composed of interconnected hexagonal microcavities and a microlens array (MLA) were used in this study, and Fig. 4a, f shows the well-stitched large-area micropatterned surfaces obtained using the MAPI process. The MAPI-based stitching was repeated four times to produce a large-area flexible mold of $200 \times 200$ mm² from small master stamp of $100 \times 100$ mm². The statistical data revealed that the angular and translational offsets in the stitched areas using the MAPI process were significantly reduced and were unperceivable to the naked eye. The measured rotational offsets ($\Delta\theta$) at the stitched areas using the MAPI process were less than 0.05° (Fig. 4e, j), as expected from the offset ranges in the simulated results (Fig. 2), which demonstrates the effectiveness of the proposed MAPI process, as compared to manual stitching.

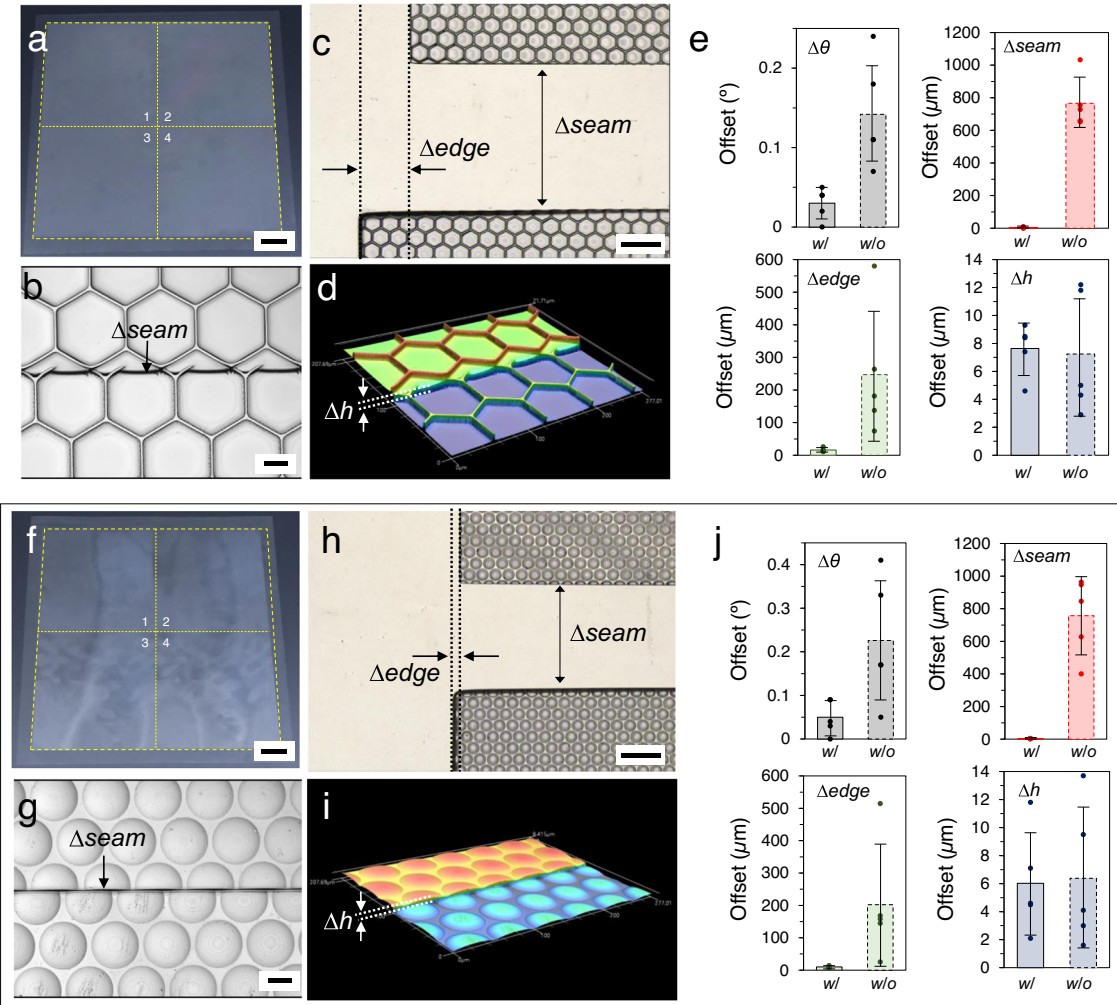

**Fig. 4 | Quasi-seamless large-area microstructured surfaces using step-and-repeat stitching based on the MAPI process (stitching sequence: 1 → 2 → 3 → 4).** Photographs of the MAPI-based stitched microstructured surfaces: **a** interconnected hexagonal microcavity and **f** MLA film. Representative 3D profile or optical images for translational (seam and edge) and height offsets: **b–d** interconnected hexagonal microcavity and **g–i** MLA films. Statistical analysis ($n = 5$) for the rotational, translational, and height offsets in stitched microstructured surfaces ($w/$: with MAPI process, $w/o$: without MAPI process): **e** interconnected hexagonal microcavity and **j** MLA films. Scale bars are 20 mm (**a, f**), 20 µm (**b, g**), and 100 µm (**c, h**). Error bars represent the standard deviations of five independent data points.

The positive effectiveness of the proposed stitching method was confirmed by further analyzing the translational offsets. The translational offsets in the stitched area were defined as $\Delta seam$ and $\Delta edge$, as shown in Fig. 4b, c and g, h, respectively. Figure 4b, g shows that the measured $\Delta seam$ was <4.5 µm, and therefore was difficult to perceive using the naked eye. However, it should be noted that a discrepancy in microstructures was found in the stitched areas because the size of the shadow mask cannot be divided by an integer multiple of the unit cell size of microstructures used in this study. Accordingly, this causes differences in the micropatterns at the border lines (that is, top, bottom, right, and left) of the shadow mask, thereby producing an inevitable discrepancy in the stitched patterns. Interestingly, the measured $\Delta edge$ exhibited relatively larger values than those of the $\Delta seam$ (Fig. 4e, j), which is probably due to the sequence of the MAPI stitching process compensating for the order of $\Delta\theta$, $\Delta edge$, and $\Delta seam$, as shown in Fig. 3. Therefore, we expected that the translational offsets observed in the MAPI process can be improved either through further process optimization or by designing the mold patterns of the MAPI stamp, as shown in the Supplementary Figs. 7–8. Meanwhile, in contrast to the MAPI process, the measured $\Delta seam$ in manual stitching was on the order of hundreds of micrometers owing to the initial random orientation state and was significantly larger than the measured $\Delta edge$,

which was also on the order of hundreds of micrometers. This is due to the empirical alignment process compensating for the order of $\Delta seam$ and $\Delta edge$ during manual stitching. Furthermore, no significant differences in $\Delta h$ were found between the MAPI and manual stitching processes because $\Delta h$ is mainly related to the residual layer thickness, which is determined by various parameters including the imprint pressure, resin viscosity, and initial amount of applied resin, rather than the stitching process[42]. Thus, it is expected that this $\Delta h$ issue can be addressed with an advanced resin dispensing system, further process optimization[43,44], or improved fabrication process of the MAPI stamp (Supplementary Fig. 4c).

## Demonstrations and applications of the MAPI-based stitched micropatterned surfaces

Figure 5a shows the R2R-based imprinting system with quasi-seamless flexible molds stitched using the MAPI process. As key examples of large-area functional surfaces continuously replicated by R2R NIL, hydrophobic structure with an interconnected hexagonal microcavity and an optical film with MLA (Fig. 4) were fabricated to investigate the effect of the seam area on their functionality. These quasi-seamless flexible molds were wrapped around a steel roll and replicated functional films were continuously fabricated (Supplementary Fig. 9). Here,

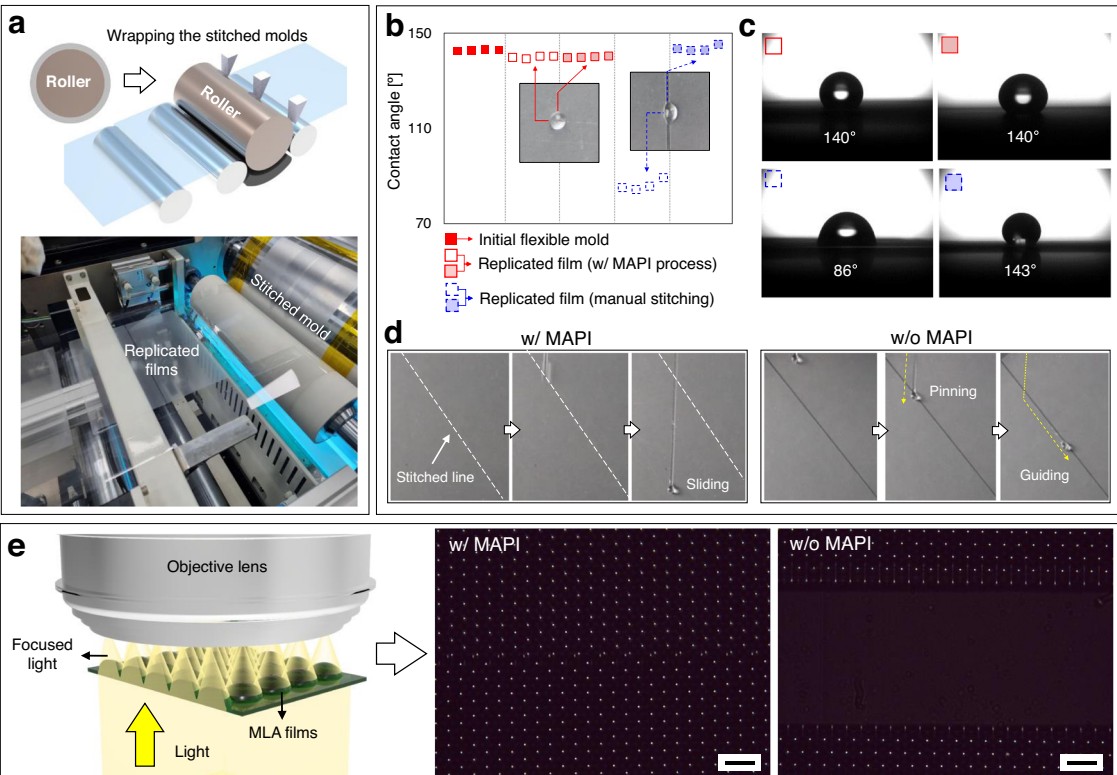

**Fig. 5 | R2R imprinting process for the continuous fabrication of large-area functional films with flexible molds formed using the MAPI and manual stitching processes. a** R2R-NIL system using a wrapped flexible mold around the roller. **b**, **c** The measured contact angle on the initial flexible mold and their replicated films fabricated using a multiple-stitched flexible mold (*n* = 3 in each area). **d** The wetting and pinning property of sliding droplets in the seam areas. **e** The light focusing characteristic of the replicated MLA films in the seam areas. Scale bar: 100 μm.

functional films with and without seam areas were prepared from two types of multiple-stitched flexible molds using the MAPI and manual stitching processes. The clear seam areas formed using the manual stitching process, as compared to the quasi-seamlessness using the MAPI process, significantly contributed to the change in the wetting or optical properties of the imprinted functional surfaces. Figure 5b–c shows the wetting behavior of the tessellated imprinted surfaces as a result of the seam area that was created by each stitching process. Because of the air pocket effect and the small contact area between the droplet and constituent solid material[45], the initial flexible mold for the interconnected microcavity exhibited superior liquid repellency, as depicted in Fig. 5b, c and Supplementary Fig. 10. The contact angle for the water droplet at the stitched area of the replicated film was measured to approximately 140° in this study. However, it can be improved to above 150° to exhibit superhydrophobicity by adjusting the pattern geometry of interconnected microcavity as shown in Supplementary Fig. 11. Moreover, the liquid contact angle at the stitched area of the replicated film, which was imprinted using the MAPI-based stitched flexible mold, exhibited a clear isotropic wetting property due to the quasi-seamlessness of the stitching process (Fig. 5b, c and Supplementary Fig. 10). In contrast, the all test liquid droplets tended to pin in the clear seam areas without microstructures of the replicated films fabricated using the manually-stitched flexible mold, thereby exhibiting anisotropic wetting behavior (Fig. 5b, c and Supplementary Fig. 10) and transition of the sliding direction (Fig. 5d). Meanwhile, the inevitable Δ*h* in the stitched area may affect to the wetting behavior of the liquid droplets as shown in Fig. 5b, c and Supplementary Fig. 10. Because of the large difference in scale between the droplet size of around millimeters and the Δ*h* of around micrometers, the effect of the Δ*h* may be negligible for liquids with high surface energy, such as water (72.8 mN/m) and glycerin (63.0 mN/m). However, the Δ*h* can

affect the wetting behavior for liquids with low surface energy, such as olive oil (38.4 mN/m). As clearly shown in Fig. 5 and Supplementary Fig. 10, the contact angle for the water and glycerin in the stitched areas with the MAPI process remained similar to those observed in the original mold patterns but the contact angle for the olive oil was slightly decreased, which might be due to the effect of the Δ*h*. Furthermore, it should be noted that the Δ*h* in the current configuration could be reduced or minimized by further optimizing the fabrication process of the MAPI stamp as presented in Supplementary Fig. 4c. In addition, a visible seam area can affect the optical properties of the replicated MLA films. To investigate the effect of the seam area on the optical property, focused light intensity was measured using an optical microscope (Olympus BX51). Under the transmission mode of the microscope, the replicated MLA film was illuminated with white light from the bottom of the sample holder stage. Images of the focusing plane were recorded from the top of the sample using an objective lens. Figure 5e shows the 2D focused light intensity distribution of the replicated MLA sheet produced by R2R-NIL using flexible molds stitched by the proposed MAPI and manual stitching processes. Although the stitched line created using the MAPI process could be clearly identified, no significant effect on the overall optical properties of the replicated MLA film was observed. In contrast, there was a significant loss of the MLA focusing property in the visible seam area created using the manual stitching process. These results demonstrate that our MAPI process can improve the reliability of its unique function in large-area microstructured surfaces by suppressing the physical seam area.

In conclusion, an MAPI process that enables large-area flexible imprinting molds by stitching the small micropatterned stamp without visible seam area was developed in this study. The main mechanism of MAPI is to utilize the rotational invariant property in the Fourier

spectral analysis of the moiré pattern produced by two superposed identical patterns, thus enabling the replacement of the conventional alignment mark. Through the MAPI process described herein, the rotational and translational offsets were extracted and subsequently adjusted to be minimized using precise motorized stages. In addition, the proposed MAPI process was confirmed to be effective for the stitching of quasi-periodic structured surfaces and periodic complex structures, which suggests that our approach may be applicable to metasurfaces composed of a periodic or aperiodic arrangement of meta-atoms[46] (Supplementary Fig. 2). Moreover, as examples of the practical applications of large-area and quasi-seamless imprinting molds and functional surfaces, robust hydrophobic surfaces and optical sheets by the multiple stitching of small micropatterned stamps using the MAPI process were successfully demonstrated. Although the MAPI approach in the present implementation is simple and effective, the main limitation to the alignment performance at the microscale arises from the optical system of the monitoring system, because the obtainable minimum feature size that can be analyzed in the MAPI process relies on the optical magnification of the objective lens in the vision-based monitoring system. However, we anticipate that the alignment performance of the vision system can be further optimized and improved by incorporating a higher-magnification objective lens or integrating novel moiré methodologies, such as the use of phase change of moiré fringes and sampling moiré method[47,48]. Another consideration is that the effectiveness of this MAPI approach was demonstrated in this study using only periodic microstructures. However, the authors believe that this does not imply a fundamental limit in the performance of this approach because the rotational offset in aperiodic structures could be also analyzed using Fourier spectral analysis (Supplementary Fig. 2). Although an increased uncertainty of the rotational offset due to complex spectral components and a complicated analysis of the inevitable translational offsets in aperiodic structures should be considered, further studies are suggested for process optimization, such as implementing machine learning-based algorithms[49], in various patterns including periodic, aperiodic, or heterogeneous arrangement to fully utilize the potential of the proposed MAPI process. This MAPI strategy can be used to expand the capability of fabricating various large-area and quasi-seamless functional patterned surfaces that are yet to be fully used at practical scales in mass production applications.

## Methods

### Fabrication of the MAPI stamp

The working MAPI stamp was fabricated using a UV-curable polymer (MINS 311RM resin, Changsung sheet, Korea) attached to chrome-coated glass with a rectangular-shaped transparent pattern (namely, a chrome shadow mask). The rectangular transparent area was 100 mm × 100 mm at the center of the chrome shadow mask having an area of 9 in × 9 in and a thickness of 3 mm. First, a surface molecular activator (GP083, GLASS PAINT, USA) was smoothly coated on the chrome-coated side for UV NIL. Then, the UV-curable resin was uniformly coated onto the chrome mask, and the micropatterns were imprinted using a transparent soft mold. An LED UV lamp with an irradiance of 110 mJ/cm² (LZL-FL-220220, LICHTZEN, Korea) was used to exposure the NIL process. After gently demolding the soft mold from the chrome mask, post-curing was performed using an LED UV lamp. Finally, oxygen plasma cleaning was performed for 5 min and trichloro(octadecyl)silane (Sigma-Aldrich, USA) was deposited to improve the release properties during the MAPI process. For demonstration purpose, two types of representative microstructured surfaces composed of interconnected hexagonal microcavities (wall width of 2 μm, height of 9 μm, and center-to-center distance of 70 μm) and MLA (diameter of 47 μm, sag height of 3.5 μm, and pitch of 50 μm) were used.

### R2P printing system

The R2P printing system consisted of four components (Supplementary Fig. 3). The vision system for real-time monitoring was based on two cameras (MV-KQ60G-P, CREVIS, Korea) equipped with a magnification lens (TCL 2.0X-65D-ST, CREVIS, Korea). The resolution, pixel size, and frame rate of the vision camera were 2592 × 1944, 2.2 × 2.2 μm, and 14 fps, respectively. The working distance and numerical aperture of the magnification lens were 65 mm and 0.065, respectively. High-precision rotational (rotation angle accuracy: ±0.1° and moving range: ±5°) and translational stages (accuracy: ±0.01 mm and moving range: ±5 mm) were used in this study for the alignment process. An LED UV lamp (area: 100 mm × 100 mm, wavelength: 365 nm, irradiance: 200 mW/cm², 2 SD, Korea) was used for photocuring. To maintain the tension in the film, the winding system consisted of two winders (Anytech, Korea).

### Multiple stitching using the MAPI approach

The stitching process for the fabrication of quasi-seamless and large-area functional surfaces or R2R imprint soft molds was performed using the MAPI stamp and R2P printing system. For this process, photocurable resin (MINS 311RM) was dropped onto the MAPI stamp, which was mounted on the stages in the R2P system. Then, a transparent polyethylene terephthalate (PET) film was gently placed on the dropped resin and pressured to ensure conformal contact between the PET film and MAPI stamp. The overlapped micropatterns were imaged to extract the misalignment offsets using vision system. After the MAPI process based on the extracted offset values, UV light was irradiated to cure the photopolymer resin using a chrome shadow mask. Finally, the MAPI stamp was gently demolded and shifted to the next stitching area.

## Data availability

All relevant data in this study are included in the article and its Supplementary Information, and are available from the corresponding authors upon request. The source data is provided with this work as a Source data file. Source data are provided with this paper.

## Code availability

The code used in the present work is available from the corresponding authors upon request.

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

## Acknowledgements

This work has supported by the National Research Foundation of Korea(NRF) grant funded by the Korea government (MSIT) (No. 2019R1A5A8083201, 2022R1C1C1003966 and 2023R1A2C3006499). The authors also acknowledge funding from the Korea Institute of Machinery & Materials under the project (NK242E).

## Author contributions

W.Y.K. and B.W.S. built the system. W.Y.K. performed experiments and characterized stitched patterns. W.Y.K. and S.K. (Seok Kim) illustrated schematics and wrote the manuscript. S.H.L., T.G.L., and S.K. (Sin Kwon) performed the roll-to-roll imprint lithography process of stitched-functional molds. W.S.C., S.N., and N.X.F. discussed and analyzed the

results. S.K. (Seok Kim) and Y.T.C. conceived the idea and supervised the project. All authors contributed to the idea discussion and edited the manuscript.

## Competing interests

W.Y.K., B.W.S., S.K. (Seok Kim), and Y.T.C. are inventors on an invention disclosure of two patents (10-2379451KR, 23 March 2022 and 10-2440860KR, 1 September 2022) related to this work. The remaining authors declare no competing interests.
