## [Peer Review File · Nature Communications]

Quasi-seamless stitching for large-area micropatterned surfaces enabled by Fourier spectral analysis of moiré patternsREVIEWER COMMENTS

Reviewer #1 (Remarks to the Author):

The authors proposed an alignment method by using Fourier spectral analysis of moire patterns. Stitching is very important to fabricate large-area optical or functional films in nanoimprint lithography. The alignment method (Self-Pattern Alignment) seems to be convenient and suitable to minimize defects which is inevitable during step-and-repeat stitching process. To convince the importance of the work, a few issues should be addressed before publication.

1. The SPA should be compared to the method with alignment marks, which is the conventional method of photolithography. If it is possible, the SPA method can be used for stitching process in photolithography as well. The authors compared the results with the conventional manual stitching process in Figure 4 and supplementary Figure 4, and it is necessary to explain the conventional manual stitching process in detail.

2. The quasi-seamless stitching process is useful in optical films because the defects cannot be seen in naked eyes. However, the surface should have a height difference, which is inevitable, and it would be not good at seamless wetting of a water droplet. For example, the authors demonstrated sliding a water droplet. What is the results when oil is wetted on the stitched surfaces?

Reviewer #2 (Remarks to the Author):

The manuscript demonstrates a moire pattern based roll-to-roll nanoimprinting technique which offers a seamless stitching (alignment error free). The quality of this stitching process depends on the computer vision and the compensating stages.

This is a promising engineering development and suitable for a manufacturing/fabrication related journal.

Reviewer #3 (Remarks to the Author):

The author proposed a self-pattern alignment (SPA) method inspired by moiré engineering to fabricate scalable functional surfaces and flexible imprint molds with quasi-seamless and alignment mark-free patterning, which is based on the Fourier spectral analysis of moiré patterns formed by superposed identical patterns. This method is innovative and valuable, and is of importance for the fabrication of large-scale flexible surface with micro-structures. However, before published in Nature communication, this article needs to be revised.

1. The MLA films shown in Fig. 4 cannot well demonstrate the innovation of this work. The problem of the seam errors is not completely solved, and whether the processing quality can be further improved through optimization.

2. Can the imprinting patterns in Fig. 4 be perfectly stitched through mold pattern designing?

3. The contact angles of these replicated films are not as high as 150°, and is it related to the rolling pressure, speed, and temperature? Successful fabrication of the replicated films with contact angles above 150° will expand its application range.

4. In this work, a transparent polyethylene terephthalate was chosen as the template. Can the material with lower hardness be only selected as the processing materials. Please list a few examples. Can metal materials be processed by this method?

5. In addition to improving the quality of processing gap, how does this method have any effect on surface quality? For example, mechanical properties.

Reviewer #4 (Remarks to the Author):

This manuscript presents a quasi-seamless stitching for fabricating a flexible mold stamp with large-area micropatterned surfaces based on the roll-to-roll nanoimprint lithography technique. As the authors developed, the self-pattern alignment (SPA) method was proposed to fabricate flexible imprint molds with alignment mark-free patterning. The critical point of this method is to use the Fourier spectral analysis of the moiré pattern generated by superposed identical patterns. All the figures and graphs in the paper are well done and very easy to read and understand. It is very impressive that the joints (seam) of the actual films, as shown in Supplementary Fig. 4 and Fig. 5, produced using the proposed method are so well manufactured that they cannot be visually confirmed. The fabrication results are sound and I am happy to recommend this research for publication after several technical points can be clarified by the authors.

Q1. What is the analysis speed of the proposed method during the alignment process? This point is not clear in the current paper.

Q2. According to the rotational invariant property of FT, the measured rotation angle is a function of the input rotation angle, as shown in Fig. 2(b) or Supplementary Fig. 1(b). What kind of formula is used to calculate the rotation angle needs to be clarified since the Fourier spectrum distribution also changes when the pattern of the regularity pattern changes. At least, Supplementary information should describe the detailed calculation principles. Is it a fully or semi-automatic calculation? In the current manuscript, the authors only mentioned three references and three rotated diffraction binary spots at the farthest point to be used to calculate the angular rotation, in lines 129-130.

Q3. How exactly is the absolute alignment of the amount of deviation Δx and Δy from the parallel shift of the repeat pattern in the x and y directions performed? Or does it matter if the multiple integer period is off? From lines 100-101, "The spatial offset for translational misalignment can be extracted by analyzing the shifted pixels between two identical periodic patterns", the reviewer supposes that the accuracy of the alignment of the parallel movement is about a pixel.

Q4. The amount of misalignment in parallel movement is detected from image processing, and alignment is performed by detecting the amount of misalignment. For (arbitrary) regular patterns, misalignment can be measured with a high accuracy of 1/1000 of the grid pitch from the phase change of moiré fringes [Ref-1] obtained by another image processing (sampling moiré method [Ref-2]), i.e., the down-sampling and intensity interpolation. I believe that such a method can be introduced in this study to achieve alignment with higher accuracy. The authors would appreciate comments on this point, as a previous work related to the recent moiré methodology.

[Ref-1] Optics Express, 22-8, 9693-9706 (2014)

[Ref-2] Experimental Mechanics, 50-4, 501-508 (2010)

Q5. To what extent can the limitations of the proposed method cope with more complex regular patterns used in various metamaterials? Because with more complex regularity patterns, the Fourier spectrum (reciprocal lattice in the k-space) will also have a rather complex distribution (a large number of Fourier spectra will appear in the distribution), making it challenging to identify angular deviations automatically and robustly.

Minor concerns include:

Abstract, line 20: There needs to be a clear and rename in a slightly different way because the self-pattern alignment (SPA) method is also often used in another SPA (spatial phase analysis) method or Software process assessment (SPA) popularly.

Abstract, line 20: "Moiré technique" seems more natural than "Moiré engineering"; the word Engineering feels uncomfortable to describe a method.

Line 127: What's the meaning of the "FTed" image? The spectral image after Fourier transform (FT)?

Dear Reviewers

Thank you very much for reviewing our manuscript and providing valuable feedback on our work. We are very pleased to see supportive comments such as “*This method is innovative and valuable*”, “*It is very impressive*”. We also appreciate the reviewers’ valuable comments for further improvements. As a result, we have considered the remarks and made all necessary changes to fully address the comments. The following contains our responses to the comments. **Referees comments are produced verbatim with our response in blue texts.**

Reviewer #1

The authors proposed an alignment method by using Fourier spectral analysis of moiré patterns. Stitching is very important to fabricate large-area optical or functional films in nanoimprint lithography. The alignment method (Self-Pattern Alignment) seems to be convenient and suitable to minimize defects which is inevitable during step-and-repeat stitching process. To convince the importance of the work, a few issues should be addressed before publication.

We are grateful for the comments and insightful question raised by the reviewer. We have tried to clarify the point raised by the reviewer and reflected them in the revised manuscript.

1. The SPA should be compared to the method with alignment marks, which is the conventional method of photolithography. If it is possible, the SPA method can be used for stitching process in photolithography as well. The authors compared the results with the conventional manual stitching process in Figure 4 and supplementary Figure 4, and it is necessary to explain the conventional manual stitching process in detail.

[Response] We thank the reviewer for these comments. Because the proposed SPA is an alignment process that utilizes ‘the patterns themselves’ without marks for quasi-seamless stitching of large-area patterned substrates, unlike the conventional method that uses marks, thus this proposed method could be applied to alignment process of multiscale patterns in the conventional lithographic methods including photolithography and nanoimprint lithography. In the alignment methods that rely on the align marks, it can be challenging to fabricate the marks precisely on the target position, especially in applications that requires seamless large-area substrates with a width over 600 mm because the alignment mark should be formed on the outside of the intended patterned area. Due to this challenge associated with fabricating marks, we think that it may be difficult to directly compare the alignment methods using the align marks with the proposed method.

In addition, the use of the term ‘conventional’ in the initial manuscript may lead to confusion and difficulties in understanding the manuscript. The term ‘conventional’ was intended to refer to the alignment process that does not use the proposed method in the initial manuscript, rather than the conventional alignment method with align marks. To prevent this confusion and misunderstanding, we revised the term ‘conventional manual stitching’ and ‘conventional process’ in the initial manuscript to ‘manual stitching’ in the revised manuscript.

For better understanding, the term ‘self-pattern alignment (SPA)’ has been further revised to ‘mark-less alignment by pattern itself (MAPI)’, which authors believe is more appropriate and intuitive than the original term. Thus, this change has been made in the overall revised manuscript.

In the MAPI process (**Supplementary Fig. 3a**), the objective is to analyze the overlapped area between the imprinted pattern and MAPI stamp pattern using the vision system (5). After the initial UV NIL procedure, the stage is moved to the next patterning area using the stage translation system (2). The resin is then filled between the MAPI stamp and the film (4). The vision system (5) analyzes the overlapped pattern area in real-time to measure the alignment error, and the MAPI stamp and film is precisely aligned (Fig. 1b) using the precision alignment system (1). The cured resin is demolded after

applying UV exposure. The MAPI process is repeated to create large-area patterns (**Supplementary Fig. 3b**). In contrast, the manual stitching process only roughly aligns the patterns using the stage translation system (2) and the winding system (3) without precise alignment via the vision system (5). We have added the description of MAPI in the revised supplementary information on page 3.

Supplementary Figure.3 Systems for the MAPI process. **a** Schematic of the roll-to-plate printing system used to fabricate quasi-seamless large-area microstructured surfaces using the MAPI process, which was comprised of the precise motorized stages and vision system. **b** The representative printing result of large-area micropatterned surfaces ($200 \times 200 \text{ mm}^2$). Scale bar: 20 mm.

2. The quasi-seamless stitching process is useful in optical films because the defects cannot be seen in naked eyes. However, the surface should have a height difference, which is inevitable, and it would be not good at seamless wetting of a water droplet. For example, the authors demonstrated sliding a water droplet. What is the results when oil is wetted on the stitched surfaces?

[Response] As the reviewer mentioned, we agree that the height difference (Δh) in the seam area is inevitable and this may affect to the wetting behavior of the liquid droplets. Because of the large difference in scale between the droplet size of around millimeters and the Δh of around micrometers, the effect of the Δh may be negligible for liquids with high surface energy, such as water (72.8 mN/m) and glycerin (63.0 mN/m). However, for liquids with low surface energy, such as olive oil (38.4 mN/m), the Δh can affect the wetting behavior.

To clarify this point, we experimentally investigated the wetting behavior in the stitched areas on the various liquid droplets and have added the results in the **supplementary Figure 10**.

In the stitched areas with the MAPI process, the contact angle for the water and glycerin remained similar to those observed in the original mold patterns and the contact angle for the olive-oil was slightly decreased, which might be due to the effect of the Δh . Furthermore, we observed a clear isotropic wetting property due to the quasi-seamlessness in the stitched areas (please see the **Figure 5b-c** and **supplementary Figure 10**). However, in the manually stitched areas without the MAPI process, the apparent anisotropic wetting property was observed in all test liquid droplets because the droplets tended to pin in the clear seam areas without microstructures of the stitched areas (please see the **Figure 5** and **supplementary Figure 6**). Furthermore, we believe that the Δh could be reduced or minimized by further optimizing the fabrication process of the MAPI stamp suggested in the **supplementary Figure. 8**.

Reflecting on the reviewer's comments, we added the detailed explanations in the revised main text and the **supplementary Figure 6** in the revised supplementary information.

Supplementary Figure. 10 The measured contact angles on the initial mold and their replicated films fabricated using a multiple-stitched flexible mold. **a** Glycerin. **b** Olive oil.

Reviewer #2

The manuscript demonstrates a moire pattern based roll-to-roll nanoimprinting technique which offers a seamless stitching (alignment error free). The quality of this stitching process depends on the computer vision and the compensating stages.

This is a promising engineering development and suitable for a manufacturing/fabrication related journal.

[Response] Thank you for the comments. Our study introduces a novel alignment algorithm, called Mark-less Alignment by pattern itself (MAPI), which allows for quasi-seamless stitching in the fabrication of large-area flexible functional surfaces without the use of alignment marks. The MAPI process utilizes Fourier spectral analysis of moiré patterns of superposed patterns themselves, thus eliminating the need for the alignment marks. By using the rotational invariance property of Fourier transforms, we can extract offsets using object patterns as the alignment mark. To our knowledge, this is the first approach for aligning and stitching the patterns to producing large-area substrates without alignment marks. Considering the compatibility of the proposed MAPI process with conventional alignment methods that utilize vision systems and translation stages, we believe that it can extend their capabilities to other lithographic approaches, including photolithography. In this study, the proposed MAPI process was demonstrated using nanoimprint lithography (NIL) as a model lithographic approach, which has several challenges when fabricating large-area substrates. These challenges include:

- (1) Accurate mark placement: The alignment mark must be carefully placed to ensure that they can be properly located and aligned during the NIL process.
- (2) Mark visibility: The alignment marks must be visible and easy to locate under the microscope or other imaging system used for alignment.
- (3) Due to the challenges posed by issues (1) and (2), it is crucial to etch the alignment marks on the edge of the master stamp.

Commonly used methods for making alignment marks easily identifiable include creating circular and cross shapes [Cho, Y., et. al., *Microelectronic Engineering*, **86**, 2417-2422 (2009); Kataza, S., et. al., *Japanese Journal of Applied Physics*, **48**, 06FH21 (2009)] or a line pattern to create a moiré fringe [Shao, J., et. al., *Optical Engineering*, **47**, 113604 (2008)]. The method described here involves engraving a separate alignment mark on the edge of a master stamp and performing step and repeat nanoimprint lithography (NIL) using this stamp. However, this results in a large gap in the area where the patterns connect, which limits the usable area for applications such as functional films (e.g., hydrophobic surfaces or optical films). To address this issue, several studies have been conducted to perform step and repeat NIL without using alignment marks [Choi, J. H., et. al., *Japanese Journal of Applied Physics*, **51**, 06FJ01 (2012), Kim, J., et. al., *Micromachines*, **9**, 569 (2018)]. However, these approaches can result in significant alignment offsets because they do not involve the alignment process. While other researchers have demonstrated quasi-seamless patterns using dynamic nanoinscribing (DNI) [Ahn, S. H., et. al., *Nano letters*, **9**, 4392-4397 (2009)] and vibrational indentation patterning (VIP) [Ahn, S. H., et. al., *Advanced Functional Materials*, **23**, 4739-4744 (2013)], these approaches are limited in the maximum achievable surface width of around 50 mm.

Considering these issues, we believe that the proposed MAPI process, which enables the precise alignment and stitching of patterns without the need for alignment marks, has the potential to offer a new solution for addressing the challenges associated with producing large-area functional surfaces. Furthermore, the use of Fourier transform and moiré techniques in this study may be of interest to researchers in fields such as physics and materials science.

Reviewer #3

The author proposed a self-pattern alignment (SPA) method inspired by moiré engineering to fabricate scalable functional surfaces and flexible imprint molds with quasi-seamless and alignment mark-free patterning, which is based on the Fourier spectral analysis of moiré patterns formed by superposed identical patterns. This method is innovative and valuable, and is of importance for the fabrication of large-scale flexible surface with micro-structures. However, before published in Nature communication, this article needs to be revised.

We appreciated the reviewer's valuable comments and insightful question. We have tried to clarify the points raised by the reviewer and reflected them in the revised manuscript.

1. The MLA films shown in Fig. 4 cannot well demonstrate the innovation of this work. The problem of the seam errors is not completely solved, and whether the processing quality can be further improved through optimization.

2. Can the imprinting patterns in Fig. 4 be perfectly stitched through mold pattern designing?

[Response] We agree that the noticeable translational offsets (Δ_{seam} and Δ_{edge}) in Fig. 4 can be improved through the process optimization and mold pattern design. The translation offsets are mainly caused by a size mismatch and rotational offset between the transparent area and printed patterns on the shadow mask. The source of the offsets may be related to the alignment or registration of the different fabrication process of the chrome-coated shadow mask and printed patterns. This suggest that these issues can be addressed either through the improved mold pattern design or through the process optimization techniques.

During the fabrication process of the MAPI stamp, the resolution of mechanical rotation stage (around 0.1°) is attributed to the rotational offset between the transparent area of the chrome-coated shadow mask and mold patterns. For example, the rotational offset of 0.1° yields the translation offset of around $174 \mu\text{m}$ in the horizontal direction between both edges when considering the rectangular transparent area of $100 \text{ mm} \times 100 \text{ mm}$, as presented in the **supplementary Figure. 7** in the revised manuscript. The authors believe that a precise motorized rotation stage can help address the issue of the rotational offset in the fabrication of the MAPI stamp.

Supplementary Figure. 7 The translation offsets after the stitching process. **a** with and **b** without the rotational offset of the MAPI stamp.

As already noted in the initial manuscript, the transparent area in the shadow mask was not divided by an integer multiple of the unit cell size of the microstructures used in this study and this leads to discrepancy of the stitched patterns at the border lines. We suggest a potential solution to this issue, which is to modify or optimize the mold pattern design to better align with the ending edge's shape of the microstructure unit at the border lines of the shadow mask. The schematic described in the **supplementary Figure. 8** in the revised manuscript may provide further guidance on how to achieve this optimization. By improving the alignment between the mold patterns and chrome-coated shadow mask, it may be possible to minimize the translational offsets and improve the overall quality and accuracy of the stitched patterns in the MAPI process.

Supplementary Figure. 8 The design strategy of mold patterns in the MAPI stamp for minimizing the translational offsets. **a** Hexagonal grid pattern **b** Micro lens array pattern.

As mentioned in Reviewer#1-Question#2, it is important to consider the height offset (Δh) in addition to the translation offsets when creating the multiple-stitched patterns with the MAPI process. Given the impact of the height offset (Δh) on the overall quality of the stitched area, we suggest optimizing the fabrication process of the MAPI stamp to reduce or minimize this factor. The **supplementary Figure 4** in the revised manuscript) can provide valuable insights and suggestions for achieving this optimization.

Since the actual printed area was slightly larger than the transparent area in the chrome-coated shadow mask (**Supplementary Figure 4a** in the revised manuscript), thus occurring an inevitable overlapped area between the already imprinted patterns and the MAPI stamp patterns during the sequential stitching process. The **supplementary Figure 4b** in the revised manuscript indicates that the presence of residual resin at the border of the overlapped region is the primary cause of the Δh observed in the stitched area.

Consequently, we believe that further optimizing the fabrication process of the MAPI stamp (**Supplementary Figure 4c** in the revised manuscript) to eliminate any size difference between the printed and transparent areas may help remove or minimize the residual resin in the overlapped region, which is the main cause of the Δh measured in the stitched area.

Reflecting the reviewer's insightful comments, we have revised the main text and added the informative supplementary figures in the supplementary information.

Supplementary Figure. 4 Schematic of the MAPI stamp fabrication. a-b Process flow of the MAPI stamp used in this study. **c** Process flow of the improved fabrication of the MAPI stamp.

3. The contact angles of these replicated films are not as high as 150° , and is it related to the rolling pressure, speed, and temperature? Successful fabrication of the replicated films with contact angles above 150° will expand its application range.

[Response] Thank you very much for this insightful comment. The wetting behavior of materials is mainly determined by the constituent material and the corresponding solid fraction, which is dependent on the geometry of surface patterns [Kim, S. et. al., *Applied Surface Science*, **513**, 145847 (2020); Kim, D. H. et. al., *ACS Applied Materials & Interfaces*, **13**, 33618-33626 (2021)]. The micropatterned surfaces are not flat and have a certain surface roughness, and the contact angle can be determined using either the Wenzel model or the Cassie and Baxter model [Kim, S. et. al., *Applied Surface Science*, **513**, 145847 (2020)]. The Cassie-Baxter model assumes that when a droplet touches the micropatterned surface, the droplet is supported by the microstructures rather than being fully wetted inside the structures. The contact angle of liquid droplets can be determined in this model based on two factors: 1) the constituent material property, and 2) the geometrical property of microstructures, which is the liquid-solid fraction (or solid fraction), defined as the ratio of the area where the droplet touches the solid surface to the projected area. According to the Cassie-Baxter model, the wetting behavior of the microstructured surfaces can be controlled by adjusting the solid fraction, without changing the constituent material being used. In this study, we used an interconnected microcavity pattern with a

width of 2 μm and center-to-center distance of 70 μm as a model structure to showcase the feasibility of the MAPI process. The focus of this study was not to demonstrate a superhydrophobic surface with a contact angle above of 150°, but rather to demonstrate the potential of the MAPI process for creating large-area functional microcavity patterns. To support our argument and address the reviewer’s comments, we conducted experimental measurements of the contact angle by reducing the solid fraction (increasing the center-to-center distance) with the same constituent material (MINS 311RM resin, Changsung sheet, Korea) used in this study. The results are presented in the **supplementary figure 11** in the revised supplementary information and we confirmed that the contact angle can be achieved above 150° in the case of the center-to-center distance 172 μm in accordance with the Cassie-Baxter model.

Meanwhile, the process parameters including the rolling pressure, temperature, speed, or resin viscosity may be affected to the overall quality of large-area patterned substrates such as the stitching offset, uniformity, or transparency [Jacobo-Martin, A., et. al., *Scientific Reports*, **11**, 1-15 (2021); Leigeb, M., et. al., *ACS nano*, **10**, 4926-4941 (2016)]. While these fabrication parameters such as rolling pressure, speed, and temperature can locally affect wetting behavior due to variations in the uniformity and defects of the transferred substrates, they are not as critical as the constituent material and solid fraction in determining overall wetting behavior across a large-area substrate.

Accordingly, we have revised the main text and included additional comments and **Supplementary Figure 11** in the revised manuscript.

Supplementary Figure. 11 The improved hydrophobicity by controlling the geometrical parameter of the microstructured surfaces.

4. In this work, a transparent polyethylene terephthalate was chosen as the template. Can the material with lower hardness be only selected as the processing materials. Please list a few examples. Can metal materials be processed by this method?

[Response] We thank for the reviewer’s valuable comment. The MAPI process proposed in our study utilizes Fourier spectral analysis of superposed moiré patterns, which are captured by an optical imaging system to extract rotational offsets. To achieve this, at least one of the MAPI stamp and substrate being patterned need to be transparent for the moiré patterns to be captured using an optical imaging system. Therefore, the MAPI process can be successfully employed if at least one of the MAPI stamp and substrate being patterned is transparent, allowing the optical imaging system to capture the moiré patterns. For example, in the MAPI process, opaque substrates such as metal materials can be used with a transparent MAPI stamp, which can be made from materials such as soda-lime glass or quartz.

5. In addition to improving the quality of processing gap, how does this method have any effect on

surface quality? For example, mechanical properties.

[Response] The proposed MAPI process can significantly minimize the seam in the stitched area by Fourier spectral analysis-based alignment without requiring special alignment marks, and it is unlikely to have a significant effect on mechanical properties, such as strength and rigidity. However, as discussed in the initial manuscript (**Fig. 5**), the quasi-seamlessness in the stitched area can have a significant impact on surface functionalities, including optical and wetting properties. We anticipate that the large-area functional surfaces produced using the MAPI process are expected to find a wide range of applications, such as self-cleaning surfaces, biological applications such as microneedles and cell cultures, and optical devices such as organic light-emitting diodes (OLEDs) and liquid-crystal displays (LCDs) that use polarizer patterns.

Reviewer #4

This manuscript presents a quasi-seamless stitching for fabricating a flexible mold stamp with large-area micropatterned surfaces based on the roll-to-roll nanoimprint lithography technique. As the authors developed, the self-pattern alignment (SPA) method was proposed to fabricate flexible imprint molds with alignment mark-free patterning. The critical point of this method is to use the Fourier spectral analysis of the moiré pattern generated by superposed identical patterns. All the figures and graphs in the paper are well done and very easy to read and understand. It is very impressive that the joints (seam) of the actual films, as shown in Supplementary Fig. 4 and Fig. 5, produced using the proposed method are so well manufactured that they cannot be visually confirmed. The fabrication results are sound and I am happy to recommend this research for publication after several technical points can be clarified by the authors.

The authors are grateful for the supportive comments and an insightful question raised by the reviewer. We have tried to clarify the point raised by the reviewer and reflected them in the revised manuscript.

1. What is the analysis speed of the proposed method during the alignment process? This point is not clear in the current paper.

[Response] We appreciate the reviewer's comment on pointing out the missing details in our description of the MAPI process in the initial manuscript.

The MAPI process involves the following steps:

- (Step.1) capturing the moiré patterns in the overlapped patterns using the vision system
- (Step.2) performing the Fast Fourier transform (FFT) of the captured moiré patterns to extract the bright node points in the k -space
- (Step.3) binarizing the Fourier transformed images to clarify these specific node points
- (Step.4) calculating the rotational offset in post-processed images
- (Step.5) compensating for the rotational offset
- (Step.6) capturing the overlapped patterns after compensating for the rotational offset
- (Step.7) calculating the translation offsets using vision-based shifted pixel analyses
- (Step.8) compensating for the translational offsets

The low computational load of the FFT algorithm and vision-based shifted pixel analysis enables efficient extraction of the rotational and translational offsets from the moiré pattern in the overlapped patterns in theoretically almost real-time, from step.1 to step.4 and from step.6 to step.8.

To reflect the reviewer's comment and clarify this point, we have added more detailed descriptions in the **supplementary Figure. 3**.

2. According to the rotational invariant property of FT, the measured rotation angle is a function of the input rotation angle, as shown in Fig. 2(b) or Supplementary Fig. 1(b). What kind of formula is used to calculate the rotation angle needs to be clarified since the Fourier spectrum distribution also changes when the pattern of the regularity pattern changes. At least, Supplementary information should describe the detailed calculation principles. Is it a fully or semi-automatic calculation? In the current manuscript, the authors only mentioned three references and three rotated diffraction binary spots at the farthest point to be used to calculate the angular rotation, in lines 129-130.

[Response] Thank you for your comments. We agree that the measured rotation angle is a linear function of the input rotation angle ($\theta_M = \theta_R$) based on the rotational invariant property of Fourier transform. To assess the estimation of rotational offset on our method, we analyzed the rotational offset for angles below 10° in **Fig. 2b** and **Supplementary Fig. 2b**, considering realistic conditions in actual applications.

In response to the reviewer's comment, the following is a detailed procedure to calculate the estimated rotation angle in the Fourier transformed images. As mentioned by the reviewer, we are aware that the

spectrum distribution of the Fourier transformed images can be influenced by various factors, including the degree of periodicity, complexity, and shapes of the constituent patterns, as well as the pixel resolution of the captured image by the vision system. These factors can significantly affect the accuracy of the calculation of the rotational offset.

To minimize the uncertainty stemming from image pixel resolution, we utilized the three diffraction spot groups farthest from the center position in the binarized images (please see **Supplementary Fig. 1a**). In each of these diffraction spot groups, the highest index values were selected as representative points, thus allowing to determine the reciprocal lattice vectors a_1 and a_2 , as indicated in **Supplementary Fig. 1b**. By calculating of the angle between the reciprocal lattice vectors, we were able to estimate the rotational offset in the Fourier transformed images (**Supplementary Fig. 1c**), resulting in the corresponding rotational offset of the overlapped patterns in the real space due to the rotational invariant properties. This type of principle used for calculating the rotational angle in this work has been widely utilized to interpret the twist angle of bilayer graphene (BLG) as moiré patterns resulting from misorientations between graphene layers [Lu, Chun-Chieh, et. al., *ACS nano*, **7**, 2587-2594 (2013); Hou, Yuan, et. al., *ACS Applied Materials & Interfaces*, **12**, 40958-40967 (2020)].

Additionally, the uncertainty in estimating the rotational offset in overlapped patterns is affected by the geometrical factors of the constituent patterns, such as the degree of periodicity, complexity, and shapes. As shown in **Supplementary Fig. 2**, we speculate that the slightly increased uncertainty observed in the aperiodic pattern of Penrose tiling may be attributed to the increased diffraction component of spatial frequency spectrum resulting from the aperiodicity of the constituent patterns.

Therefore, we also plan to implement a machine learning-based algorithm in the near future to improve the accuracy of estimating the rotational offset of overlapped patterns using the MAPI process [Solis-Fernandez, P., et. al., *ACS Applied Nano Materials*, **5**, 1356-1366 (2022)].

To reflect the reviewer's comment, we have added more detailed descriptions and **Supplementary Figure 1** in the revised supplementary information.

Supplementary Figure. 1 The calculation procedure for estimating for the rotation angle between the reciprocal lattice vectors in the Fourier transformed images. **a** Extraction of three diffraction spot groups farthest from the center position in the binarized image **b** Determination of the highest index value point from diffraction spot groups **c** Calculation of the rotational offset from the reciprocal lattice vectors of a_1 and a_2

3. How exactly is the absolute alignment of the amount of deviation Δx and Δy from the parallel shift of the repeat pattern in the x and y directions performed? Or does it matter if the multiple integer period is off? From lines 100-101, “The spatial offset for translational misalignment can be extracted by analyzing the shifted pixels between two identical periodic patterns”, the reviewer supposes that the accuracy of the alignment of the parallel movement is about a pixel.

[Response] As noted by the reviewer, we agree that the theoretical accuracy of calculating the translational offsets is approximately one pixel and is dependent on the optical resolution of the vision system in the current configuration.

We would like to clarify that the analysis of translational offsets in the x and y directions is performed using the direct vision system with a magnification optic, rather than Fourier spectral analysis, which is used for analyzing rotational offset. We also believe that this accuracy of parallel movement does not indicate a fundamental limit in the performance of the vision-based approach. There are potential avenues for improvement, such as using higher magnification optics or integrating moiré methodologies as mentioned by the references in Reviewer#4-Quesiton#4 [Ri, S., et. al., *Optics Express*, **22-8**, 9693-9706 (2014); Ri, S., et. al., *Experimental Mechanics*, **50-4**, 501-508 (2010)].

Therefore, we have revised the main text and included a detailed description and **Supplementary Figure 5** in the revised supplementary information to provide additional information about the alignment procedure for translational offsets.

Supplementary Figure 5 Compensation of translational offset by extracting center points with pixel shift analysis. a Flowchart on the extraction of the center point in the overlapped patterns. **b** Examples of the extraction of the center position in various periodic structures such as triangular, square, and hexagonal patterns.

4. The amount of misalignment in parallel movement is detected from image processing, and alignment is performed by detecting the amount of misalignment. For (arbitrary) regular patterns, misalignment can be measured with a high accuracy of 1/1000 of the grid pitch from the phase change of moiré fringes [Ref-1] obtained by another image processing (sampling moiré method [Ref-2]), i.e., the down-sampling and intensity interpolation. I believe that such a method can be introduced in this study to achieve alignment with higher accuracy. The authors would appreciate comments on this point, as a

previous work related to the recent moiré methodology.

[Ref-1] Optics Express, 22-8, 9693-9706 (2014)

[Ref-2] Experimental Mechanics, 50-4, 501-508 (2010)

[Response] Thank you very much for valuable comments and sharing these informative references. As we previously noted in response to Reviewer#4-Question#3, we agree that the alignment accuracy in parallel movement can be further optimized or improved through the integration of novel moiré methodologies, such as the use of phase change of moiré fringes and sampling moiré method.

We have revised the main text and added these references to reflect the reviewer's suggestion.

5. To what extent can the limitations of the proposed method cope with more complex regular patterns used in various metamaterials? Because with more complex regularity patterns, the Fourier spectrum (reciprocal lattice in the k-space) will also have a rather complex distribution (a large number of Fourier spectra will appear in the distribution), making it challenging to identify angular deviations automatically and robustly.

[Response] The authors agree with the reviewer's comment that the corresponding Fourier spectral distribution become more complicated with an increase in the degree of periodicity, complexity, or shapes of the constituent patterns. Among those factors, we consider that the degree of periodicity is an important factor in determining the limitations of the proposed MAPI process in this study. For instance, it would be indeed challenging to extract the rotational offset in the case of fully random patterns with a degree of periodicity = 0 due to the broad distribution of Fourier spectrum, thereby making it difficult to extract meaningful information for the automatic and reliable offset calculation. However, aperiodic or periodic patterns are more prevalent in real-world applications, as they can provide certain functionalities, such as wettability and optical response. As a result, we demonstrated the estimation of rotational offset in the aperiodic patterns of Penrose tiling (degree of periodicity $\neq 1$) using the MAPI process (**Supplementary Fig. 2**) and observed an increased uncertainty compared to other types of regularized patterns and patterns of metasurfaces (degree of periodicity ~ 1). We believe that this may be attributed to the increased diffraction component of spatial frequency spectrum resulting from the aperiodicity of the constituent patterns. To address this limitation, we are planning to adapt a machine learning-based algorithm that can improve the accuracy of estimating the rotational offset of overlapped patterns [Solis-Fernandez, P. et. al., *ACS Applied Nano Materials*, **5**, 1356-1366 (2022)].

Reflecting on the reviewer's comment, we have revised the main text accordingly.

Minor concerns include:

Abstract, line 20: There needs to be a clear and rename in a slightly different way because the self-pattern alignment (SPA) method is also often used in another SPA (spatial phase analysis) method or Software process assessment (SPA) popularly.

[Response] We agree with the reviewer's comment and accordingly have changed the term 'Self-Pattern Alignment (SPA)' to 'Mark-less Alignment by Pattern Itself (MAPI)' in the overall revised manuscript.

Abstract, line 20: "Moiré technique" seems more natural than "Moiré engineering"; the word Engineering feels uncomfortable to describe a method.

[Response] We agree with the reviewer's suggestion and accordingly have changed the term 'moiré engineering' to 'moiré technique' in the revised manuscript.

Line 127: What's the meaning of the "FTed" image? The spectral image after Fourier transform (FT)?

[Response] Yes, the term 'FTed' meant that the spectral image after Fourier Transform. However, to avoid confusion, we have changed 'FTed' to 'Fourier-transformed' in the revised manuscript.

REVIEWERS' COMMENTS

Reviewer #3 (Remarks to the Author):

The authors have perfectly resolved all my questions. The manuscript can be published now.

Reviewer #4 (Remarks to the Author):

The authors have carefully revised the manuscript and addressed my concerns, and I have no further questions.

I now support the publication of this paper in Nature Communications.

Nature Communications: NCOMMS-22-46858A - Response to Reviewer Comments

Reply to reviewers

We are grateful to the reviewers for the improvement of our manuscript and suggestion for publication.

Reviewer #3

The authors have perfectly resolved all my questions. The manuscript can be published now.

[Response] Thank you for the insightful advice during review process. Reviewer's comments have significantly helped for the overall quality our scripts.

Reviewer #4

The authors have carefully revised the manuscript and addressed my concerns, and I have no further questions.

I now support the publication of this paper in Nature Communications.

[Response] Thank you for the insightful advice during review process. Reviewer's comments have significantly helped for the overall quality our scripts.